# Assessing hypoxic respiratory failure in mechanically ventilated neonates: A comparative study of oxygen saturation index and oxygenation index

**Son Bui-Binh-Bao**[1], **Dao Nguyen Thi**[2], **Linh Hoang Mai**[3], **Tam Do Ho Tinh**[1], **Thi Thanh Binh Nguyen**[1]*

**1** Department of Pediatrics, Hue University of Medicine and Pharmacy, Hue University, Hue, Thua Thien Hue Province, Vietnam, **2** Vinh Duc Hospital, Dien Ban, Quang Nam, Vietnam, **3** Pediatric Center, Hue Central Hospital, Hue, Thua Thien Hue Province, Vietnam

* nttbinh.a@huemed-univ.edu.vn, nttbinh.med@hueuni.edu.vn

**Data Availability Statement:** All relevant data are within the manuscript and its Supporting Information files.

## Abstract

### Objective

To investigate the correlation between oxygen saturation index (OSI) and oxygenation index (OI) for evaluating the blood oxygenation status in neonates with respiratory failure requiring mechanical ventilation support and to assess the predictive capability of OSI in determining clinically relevant OI cutoffs.

### Methods

A prospective study was conducted on neonates who received invasive mechanical ventilation at the neonatal intensive care unit of tertiary hospital in Vietnam. Bland-Altman analysis was utilized to evaluate the agreement between OSI and OI.

### Results

A total of 123 neonates, including both term and preterm infants, were included in the study. A high agreement rate of 94.3% within the 95% limits of agreement (between OI and OSI), with a narrow similarity value of 3.3 (95% CI: -5.1 to 11.8) and high correlation coefficient (r = 0.791, p<0.001) was observed. The OSI cut-off value for predicting an OI of >15 was determined to be 7.45, with a sensitivity of 100% and a specificity of 87.4% (AUC 0.955; 95% CI: 0.922–0.989, p < 0.05). Similarly, an OSI cutoff value of 9.9 corresponded to an OI of 25, displaying a sensitivity of 100% and specificity of 87.4% (AUC 0.92). The receiver operating characteristic (ROC) curves for OSI exhibited statistically significant results (p < 0.05).

### Conclusion

The findings demonstrate a strong correlation between OSI and OI in neonates with respiratory failure. Furthermore, OSI, as a non-invasive method, can serve as a substitute for OI to evaluate the severity of hypoxic respiratory failure and lung injury in neonates.

**Funding:** The author(s) received no specific funding for this work.

**Competing interests:** The authors have declared that no competing interests exist

## Introduction

Respiratory failure is a prevalent issue and remains the primary cause of mortality during the neonatal period [1]. In neonatal intensive care units (NICUs), the management of severe respiratory failure necessitating mechanical ventilation poses significant challenges due to its complex treatment, high treatment costs, and elevated mortality rates. Additionally, prolonged severe respiratory failure can lead to neurological complications in survivors [2]. The oxygenation index (OI) serves as a valuable marker for evaluating the severity of hypoxic respiratory failure (HRF) and lung injury in ventilated infants. It is widely utilized to guide clinical decisions, such as initiating high-frequency mechanical ventilation, administering inhaled nitric oxide therapy for neonates with respiratory failure caused by primary pulmonary hypertension, and assessing the response to surfactant therapy [3–5]. Furthermore, the OI is also a predictor of mortality in neonates with HRF [6].

The calculation of OI involves the following equation: $OI = (MAP \times FiO_2 \times 100) / PaO_2$, where MAP is the mean airway pressure and $FiO_2$ is the fraction of inspired oxygen [7]. As a result, determining the OI requires obtaining a blood sample for arterial blood gas analysis through arterial puncture or catheterization. Due to this invasive nature, arterial blood gases are typically measured intermittently during the procedure. However, these procedures can lead to complications such as local hematoma, arterial spasm, arterial thrombosis, or embolism, potentially causing distal limb ischemia or even osteomyelitis, especially with multiple sample collections. Moreover, in cases of severe respiratory failure and prolonged mechanical ventilation, frequent arterial blood gas assessments contribute to early and severe anemia in neonates.

Additionally, in resource-constrained settings like Vietnam and other developing countries, availability of blood gas analyzer machines in NICUs is limited, whereas pulse oximeters ($SpO_2$) are commonly available. To overcome the limitations of invasive procedures and propose a noninvasive alternative for continuous monitoring of oxygenation impairment in mechanically ventilated neonates, the oxygen saturation index (OSI) is considered as a substitute for the OI. The formula for calculating OSI replaces $PaO_2$ with $SpO_2$ and is expressed as $OSI = (MAP \times FiO_2 \times 100) / SpO_2$. This led to the hypothesis that the OSI could serve as a viable alternative to OI for assessing the blood oxygenation status in neonates with ventilator-associated respiratory failure, providing continuous monitoring without invasive procedures.

### Objectives

To investigate the correlation between OSI with OI indices in assessing blood oxygenation status of neonates with respiratory failure requiring mechanical ventilation and to evaluate the predictive capability of OSI for clinically significant OI cutoffs.

## Material and methods

### Study design

We performed a prospective study in the NICU of the Pediatric Center, Hue Central Hospital, in Hue City, Vietnam, between June 10, 2020, and June 31, 2022. All data were prospectively collected from patients admitted to the NICU during this period.

### Subjects

Inclusion criteria comprised 123 neonates, encompassing both term and preterm infants, who presented with respiratory distress necessitating invasive mechanical ventilation. Neonates

with cyanotic congenital heart disease, hypotension, or in a state of shock were excluded from the study.

## Variables and data collection

Encompassed various parameters, including birth weight, gestational age, gender, mode and type of delivery, as well as the causes of respiratory failure among all participants.

Additionally, arterial blood gas (ABG) parameters, corresponding oxygen saturation levels, and ventilator settings (such as FiO2 and MAP) at the time of arterial sampling were recorded. Only one pair of ABG and corresponding oxygen saturation (SpO2) value was collected from each enrolled neonate. In our study, the ABG sample was drawn after the neonate had been on a ventilator for 30–60 minutes, which is a standard practice that allows for ventilator adjustments if needed. To minimize measurement errors, newborns were kept still during sampling, and no position changes, or suctioning was performed during this time. SpO2 was continuously monitored using pulse oximetry. After observing a stable SpO2 value for at least one minute, the reading was recorded. ABG samples were obtained from umbilical arterial catheter or by arterial puncture. The OI was calculated by the formula MAP × FiO2 × 100/ PaO2, while the OSI was calculated by the formula MAP × FiO2 × 100 / SpO2. ABG analysis was performed using the Medica Corporation Cat. No. 6601-E system (USA). SpO2 was measured from the post-ductal site (left hand or both feet) if ABG was collected from the umbilical arterial catheter; and at the right wrist or hand using the Vismo PVM 2701 device (Nihon Kohden Corporation, Japan) if ABG was collected from the right radial or brachial artery.

## Statistical analysis

Pearson correlation was used to determine the correlation of OI and OSI, and linear regression techniques were employed to establish an equation representing the association between OSI and OI. Bland-Altman method was utilized to evaluate the agreement between OSI and OI [8]. The correlation analysis was performed on the entire cohort of paired samples. The receiver operating characteristic (ROC) curve was done to find the predictability of OSI. Sensitivity, specificity, positive predictive value (PPV), and negative predictive value (NPV) of OSI in predicting clinically relevant OI cutoffs (OI>15 and OI>25) were determined. In general, OI cutoffs are widely used for categorizing the severity of HRF in neonatal clinical practice. OI cutoffs are: ≤15: Mild HRF 16–25: Moderate HRF 26–40: Severe HRF 40: Very severe HRF. None of the neonates presented with very severe condition (OI >40). Therefore, considering both established cutoffs and the distribution of OI values within our data, these specific cutoffs (>15 and >25) were chosen for data analysis. Statistical Package for Social Sciences version 20.0 was utilized to perform all statistical analyses. A p-value of less than 0.05 was considered statistically significant.

## Ethical statements

This study was conducted with the ethical approval of the Institutional Review Board (IRB) at the University of Medicine and Pharmacy, Hue University (Approval No H2020/127). Before enrolling the neonates in the study, written informed consent about the study, and used data in the medical records were obtained from their parents.

## Results

For this study, we enrolled neonates with severe respiratory failure requiring invasive ventilation, all admitted to the NICU. A total of 123 neonates, including both term and preterm

**Table 1. General characteristics of study population.**

| VARIABLES | | N = 123 | % |
|---|---|---|---|
| Gender | Male | 77 | 62.6 |
| | Female | 46 | 37.4 |
| Gestational age (weeks) | <28 | 14 | 11.4 |
| | 28-<32 | 19 | 15.5 |
| | 32-<34 | 17 | 13.8 |
| | 34-<37 | 25 | 20.3 |
| | 37-<42 | 48 | 39.0 |
| | $\geq$42 | 0 | 0.0 |
| | Median (25th -75th) | 35 (31–38) | |
| Birth weight (grams) | < 1500 | 24 | 27.6 |
| | 1500-<2500 | 33 | 26.8 |
| | >2500 | 56 | 45.6 |
| | Median (25th -75th) | 2300 (1360–3000) | |
| Types of delivery | Vaginal | 44 | 35.8 |
| | Caesarean section | 79 | 64.2 |
| Causes of respiratory distress | Respiratory distress syndrome | 40 | 32.5 |
| | Sepsis | 21 | 17.1 |
| | Asphyxia | 17 | 13.8 |
| | Neonatal pneumonia | 6 | 4.9 |
| | Meconium aspiration syndrome | 6 | 4.9 |
| | Pneumothorax | 5 | 4.1 |
| | Severe anemia | 3 | 2.5 |
| | Persistent pulmonary hypertension | 2 | 1.6 |
| | Gastroschisis | 2 | 1.6 |
| | Congenital esophageal atresia | 2 | 1.6 |
| | Neonatal peritonitis | 2 | 1.6 |
| | Intraventricular hemorrhage | 1 | 0.8 |
| | Others | 16 | 12.9 |
| Oxygenation index (OI) | $\leq$15 | 103 | 83.7 |
| | 15< OI$\leq$25 | 16 | 13.0 |
| | 25<OI$\leq$40 | 4 | 3.3 |
| | >40 | 0 | 0.0 |
| | Median OI | 7.3(4.4–12.5) | |
| Oxygen saturation index (OSI) | Median OSI | 5.1 (3.4–8.1) | |

infants, were recruited, with 62.6% of them being male. The most prevalent causes of respiratory failure among the neonates were respiratory distress syndrome (32.5%), sepsis (17.1%), and asphyxia (13.8%). Other contributing factors included neonatal pneumonia (4.9%), meconium aspiration syndrome (4.9%), pneumothorax (4.1%), and persistent pulmonary hypertension (1.6%). Further details regarding the basic information of the study group were shown in Table 1. Neonates had a median gestational age of 35 weeks (range: 31–38 weeks) and median birth weight of 2300 grams (range: 1360–3000 grams). In terms of oxygenation indices, the median OI was 7.3 (range: 4.4–12.5), while the median OSI was 5.1 (range: 3.4–8.1). Among the neonates, 103 (83.7%) had an OI of $\leq$15, 16 (13.0%) had an OI between >15 and $\leq$25, and 4 (3.3%) had an OI between >25 and $\leq$40. Notably, none of the neonates exhibited a very severe condition with an OI >40. Fig 1 presents the correlation between OI and OSI, demonstrating a significant positive correlation among the neonates with an r-value of 0.791

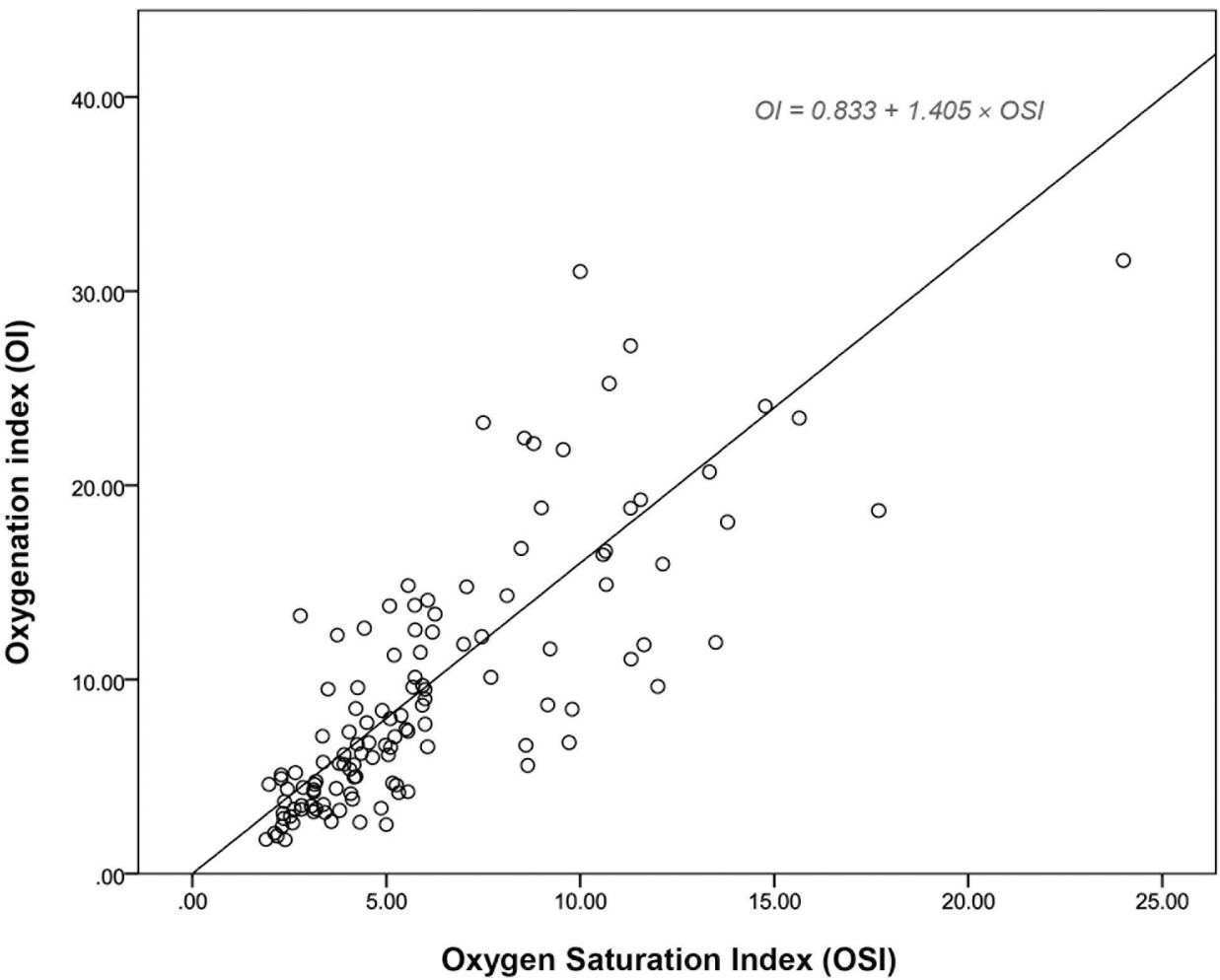

**Fig 1. Linear correlation between Oxygen Saturation Index (OSI) and Oxygenation Index (OI).**

($p < 0.05$). The regression equation $OI = 0.833 + 1.405 \times OSI$ indicates a strong linear association between OI and OSI.

Furthermore, the compatibility of OSI and OI was assessed using Bland-Altman method, as depicted in Fig 2. The Bland-Altman diagram visually represents the level of agreement between the OI and OSI methods in evaluating respiratory failure severity. The analysis demonstrated a high agreement rate of 94.3% within the 95% limits of agreement, with a narrow similarity limit value of 3.3 (95% CI: -5.1 to 11.8). Only a minor difference of 5.7% fell outside the 5% limits of the agreement.

Considering the variation in fetal hemoglobin levels due to age, we further investigated the compatibility of OSI and OI in preterm and term neonates independently (see S1 and S2 Figs). In the preterm group, the mean difference between OI and OSI was 3.5, with a 95% confidence interval ranging from -5.4 to 12.4. In the term group, the mean difference was 3.0, with a 95% confidence interval of -4.7 to 10.8. There was a high degree of agreement between OI and OSI in 2 groups of mean values, with 94.7% agreement in the preterm group and 93.7% agreement in the term group.

Moreover, the predictive ability of OSI for OI values above 15 and 25, indicating moderate and severe disease, respectively, was assessed. The cut-off value of OSI to predict an OI value

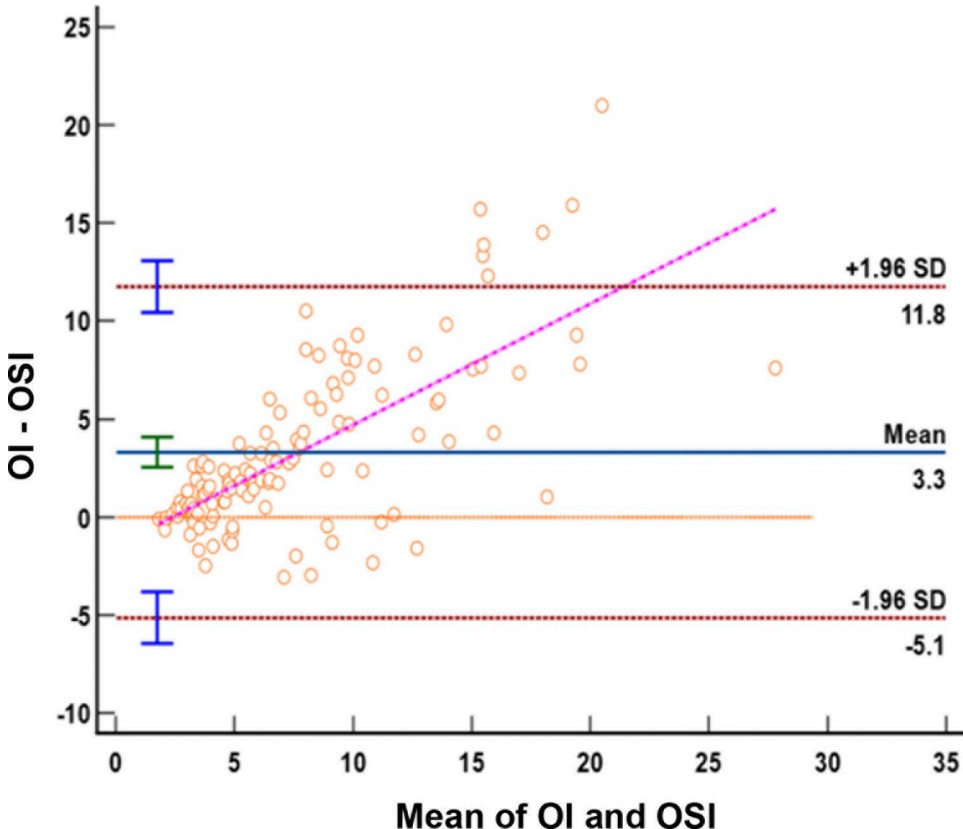

**Fig 2. Bland-Altman plot for assessing the agreement between oxygen saturation index (OSI) and oxygenation index (OI).** The diagonal line represents the slope of the relationship indicating a significant bias, $p < 0.0001$.

above 15 was determined to be 7.45, with a sensitivity of 100% and a specificity of 87.4% (AUC 0.955; 95% CI: 0.922–0.989, p<0.05). Similarly, the cut-off value of OSI corresponding to an OI of 25 was found to be 9.9, with maintained high sensitivity (100%) and a specificity of 87.4% (AUC 0.92). The ROC curves for OSI yielded statistically significant results (p<0.05) (Fig 3).

## Discussion

Our study revealed a strong relationship between OI and OSI for measuring oxygenation in critical ill neonates. These results are consistent with a prospective study of Sunil and Nithya (2021) conducted in similar settings that recruited both preterm and term in study group and each selected newborns only one paired OI and OSI measurement was evaluated. Sunil and Nithya conducted a study involving 50 neonates, both term and preterm, and reported a positive correlation between OSI and OI (p-value < 0.01) [9]. The correlation coefficient (r) was found to be 0.727, which aligns with the findings of our study. The AUC was performed to find the overall accuracy of OSI as a diagnostic test for evaluating the severity of respiratory failure. In this study, the AUC was 0.912, indicating a good overall accuracy of the test. These findings suggest that OSI can be used with good accuracy as a substitute for OI in assessing the severity of respiratory failure in neonates [9]. Besides, Devleta et al. (2018) also observed a correlation between mean values of OSI and OI in their study involving 101 ventilated neonates those similar mean of gestational age with participants our study, with a Pearson's coefficient of 0.76 (p < 0.0001, 95% CI = 0.66–0.83) [10]. Similarly, Rawat et al. (2015) conducted a study

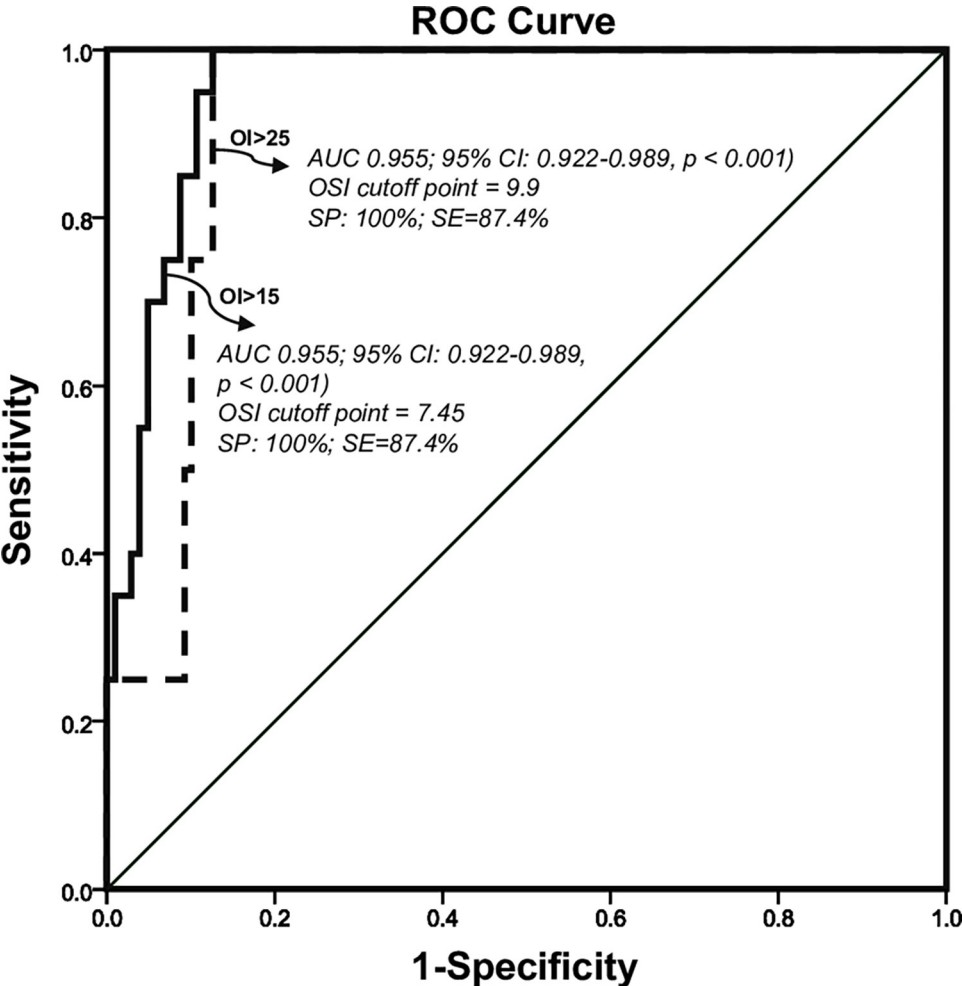

**Fig 3. The receiver operator characteristic curve for OSI in the prediction of OI.** (SP: Specificity; SE: Sensitivity); oxygen saturation index (OSI) and oxygenation index (OI).

and reported a correlation coefficient of 0.952 between mean values of OSI and OI in neonates [7]. Muniraman et al. and Maneenil et al. reported the strong correlation between the OI and OSI with $r = 0.89$, and $r = 0.90$, respectively. Unlike many previous studies on this topic, which were mainly retrospective (e.g., Devleta et al., Rawat et al., Muniraman et al., Maneenil et al.) [7, 10–12], our study is a prospective design. This approach guarantees meticulous data collection, especially in ensuring accurate pairing of SpO2 measurements with corresponding arterial blood gas samples. Furthermore, analyzing only one SpO2 and ABG pair per neonate fosters consistency in our study results compared to investigations that involve multiple measurements from the same neonates (Muniraman et al., Maneenil et al.) [11, 12]. This decision is grounded in the fact that individual hemoglobin concentration can influence SpO2 readings. For instance, Muniraman et al. employed a median of 5 (range: 3–9) samples per patient [11], while Maneenil et al. utilized a median of 8 (range: 6–12) measurements [12].

To further assess the agreement between OSI and OI, we conducted Bland-Altman analyses using mean values. Our results demonstrated that the average difference between OI and OSI mostly fell within the similarity limit, accounting for 94.3% of cases. This indicates a good agreement between the two indices and suggests their potential clinical applicability. These results are consistent with studies done by Devleta et al. and Zadkarami et al., which also

reported agreement between OSI and OI in mean values, even for extreme values [10, 13]. In summary, our study provides evidence that the noninvasive OSI, calculated by substituting SpO2 obtained by pulse oximetry for PaO2, can be effectively used as a substitute for the invasive OI in assessing the severity of respiratory failure in neonates. These findings contribute to the literature supporting the clinical utility of OSI as a reliable alternative to OI in neonatal respiratory care.

We conducted an analysis to determine the optimal values of OSI corresponding to moderate and severe lung injury in neonates with respiratory failure. Our results indicated that an OSI value of 7.45 was the optimal cut-off level for diagnosing moderate respiratory failure, which corresponded to an OI threshold of 15. Additionally, an OSI value of 9.9 corresponded to an OI of 25. These cut-off values demonstrated high sensitivity, specificity, and negative predictive value, indicating the accuracy of OSI in defining respiratory failure severity in neonates. The AUC was 0.955 for the moderate respiratory failure cut-off and 0.92 for the severe respiratory failure cut-off, further validating the effectiveness of OSI as a diagnostic tool. Our findings align with previous studies that have investigated OSI cut-off values for predicting severe respiratory failure [7, 14]. For instance, Rawat et al. reported an OSI cut-off of >7.5 corresponding to an OI > 15, with a sensitivity of 83% and specificity of 100% [7]. Similarly, Doreswamy SM et al. found that an OSI value with a sensitivity of 100% and specificity of 93.7% accurately predicted an OI value higher than 15 [14]. These consistent results reinforce the potential utility of OSI in assessing the severity of respiratory failure in neonates. OSI only requires pulse oximetry, which is widely accessible and available compared to the arterial blood gas analyzers needed for OI. Therefore, OSI has distinct advantage and become the preferred choice in resource-constrained setting where ABG analysis is limited or unavailable. Furthermore, even without access to advanced interventions like nitric oxide, high-frequency ventilation, or ECMO, higher OSI cutpoints can remain valuable in resource-constrained settings. They can help identify neonates at high risk of developing severe conditions, prompting closer monitoring and essential supportive care in treatment like oxygen therapy, infection control, and even referral to higher-level hospitals when possible. While these interventions may be less advanced, they can still significantly improve patient outcomes.

The present study has some limitations. First, the data collection occurred at a single tertiary-level hospital in central Vietnam. While this limits the sample size and generalizability of the findings, it is important to note that this is the largest tertiary-level hospital in the region, specializing in the care of critically ill neonates. Additionally, it is the third largest hospital in Vietnam. Therefore, despite the sample size limitation, the data offers valuable insights as a pioneer study in this topic of research within Vietnam. Second, the relationship between PaO2 and SpO2 can be affected by several factors, including pH, temperature, and PaCO2. These factors can cause a shift in the oxygen-hemoglobin dissociation curve, potentially complicating the analysis of the association between OSI and OI.

In conclusion, our study demonstrates a strong correlation between OSI and OI in neonates with HRF, indicating that OSI can effectively define the severity of lung injury in mechanically ventilated neonates. We propose that the noninvasive calculation of OSI using pulse oximetry is a valuable alternative to the invasive measurement of OI through arterial puncture. By adopting OSI as a reliable assessment method, clinicians can enhance clinical management and decision-making for neonates with respiratory failure.

## Supporting information

**S1 Fig. Bland-Altman plot for assessing the agreement between oxygen saturation index (OSI) and oxygenation index (OI) in preterm neonates.** The diagonal line represents the

slope of the relationship indicating a significant bias, $p < 0.0001$.
(TIF)

**S2 Fig. Bland-Altman plot for assessing the agreement between oxygen saturation index (OSI) and oxygenation index (OI) in term neonates.** The diagonal line represents the slope of the relationship indicating a significant bias, $p < 0.0001$.
(TIF)

**S1 Data.**
(XLSX)

## Author Contributions

**Conceptualization:** Son Bui-Binh-Bao, Dao Nguyen Thi, Linh Hoang Mai, Tam Do Ho Tinh, Thi Thanh Binh Nguyen.

**Data curation:** Son Bui-Binh-Bao, Dao Nguyen Thi, Linh Hoang Mai, Tam Do Ho Tinh, Thi Thanh Binh Nguyen.

**Formal analysis:** Son Bui-Binh-Bao, Tam Do Ho Tinh, Thi Thanh Binh Nguyen.

**Investigation:** Son Bui-Binh-Bao, Dao Nguyen Thi, Linh Hoang Mai, Tam Do Ho Tinh, Thi Thanh Binh Nguyen.

**Methodology:** Son Bui-Binh-Bao, Dao Nguyen Thi, Linh Hoang Mai, Tam Do Ho Tinh, Thi Thanh Binh Nguyen.

**Project administration:** Son Bui-Binh-Bao, Dao Nguyen Thi, Linh Hoang Mai, Thi Thanh Binh Nguyen.

**Resources:** Son Bui-Binh-Bao, Linh Hoang Mai, Thi Thanh Binh Nguyen.

**Software:** Son Bui-Binh-Bao, Thi Thanh Binh Nguyen.

**Supervision:** Son Bui-Binh-Bao, Thi Thanh Binh Nguyen.

**Validation:** Son Bui-Binh-Bao, Thi Thanh Binh Nguyen.

**Visualization:** Son Bui-Binh-Bao, Thi Thanh Binh Nguyen.

**Writing – original draft:** Son Bui-Binh-Bao, Thi Thanh Binh Nguyen.

**Writing – review & editing:** Son Bui-Binh-Bao, Thi Thanh Binh Nguyen.

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
