## [Decision Letter · Decision Letter 0]

16 Feb 2024

PONE-D-23-41595Assessing Hypoxic Respiratory Failure in Mechanically Ventilated Neonates: A Comparative Study of Oxygen Saturation Index and Oxygenation IndexPLOS ONE

Dear Dr. Nguyen,

Thank you for submitting your manuscript to PLOS ONE. After careful consideration, we feel that it has merit but does not fully meet PLOS ONE’s publication criteria as it currently stands. Therefore, we invite you to submit a revised version of the manuscript that addresses the points raised during the review process.

We look forward to receiving your revised manuscript.

Kind regards,

David Chau

Academic Editor

PLOS ONE

Journal Requirements:

Reviewers' comments:

Reviewer's Responses to Questions

**Comments to the Author**

1. Is the manuscript technically sound, and do the data support the conclusions?

Reviewer #1: Yes

Reviewer #2: Yes

Reviewer #3: Yes

2. Has the statistical analysis been performed appropriately and rigorously? 

Reviewer #1: Yes

Reviewer #2: Yes

Reviewer #3: Yes

3. Have the authors made all data underlying the findings in their manuscript fully available?

Reviewer #1: Yes

Reviewer #2: No

Reviewer #3: Yes

4. Is the manuscript presented in an intelligible fashion and written in standard English?

Reviewer #1: Yes

Reviewer #2: Yes

Reviewer #3: Yes

5. Review Comments to the Author

Reviewer #1: Other investigators have also studied the correlation of OSI and OI in neonates, including those referenced in the manuscript, Rawat, et al., Sunit and Nithya. Similar articles not referenced include:

Muniraman, et al. (JAMA Network Open. 2019;2(3):e191179) and very recently

Maneenil, et al. (Am J Perinatol. 2024 Jan;41(2):180-186).

Given the similarities between the submitted article and the referenced and unreferenced articles, the authors need to point out if and how their results are similar or differ from others.

It is unclear how many samples were obtained from the 123 enrolled patients. Were multiple samples obtained from some of the patients (as was the case for similar studies)? If not, how was the uniquely associated blood gas sample chosen?

Were arterial samples obtained via an indwelling arterial catheter (umbilical or peripheral) or by arterial stick? This is important to know as peripheral arterial sticks are more likely to have less reliability given the pain and discomfort associated with the blood draw.

Authors state on manuscript page 3: “It [OI] is widely utilized to guide clinical decisions, such as initiating high-frequency mechanical ventilation, administering inhaled nitric oxide therapy for neonates with respiratory failure caused by primary pulmonary hypertension, and assessing the response to surfactant therapy (3, 4).” Reference 3 should be updated to the 2017 Cochrane review by the same authors and others (Cochrane Database Syst Rev. 2017 Jan 5;1(1):CD000399). It should be noted that this reference only discusses the use of OI in the decision to use inhaled nitric oxide (INO). Reference 4 not being a manuscript publication warrants a website URL link. That reference discusses the use of OI for INO, possible transfer for ECMO, and to predict adverse outcome/death. Neither reference addresses the use of OI for high-frequency ventilation or response to surfactant.

Authors state on manuscript page 4: “Neonates with cyanotic congenital heart disease, hypotension, or in a state of shock were excluded…” It may be helpful to know how hypotension or shock were defined, especially considering that many babies with significant respiratory failure may have one or both, and for whom using OSI may be even more important. No such exclusion were used by Rawat, et al., Sunil and Nithya, Muniraman et al. or Maneenil, et al.

Minor editorial comments:

References in the manuscript should not use the authors first name or initial, e.g “Hadzic Devleta et al.” should read “Devleta et al.”. When there are only two authors, the referring phrase should be “Sunil and Nithya” instead of “Sunil et al.”

One reference (Sunil and Nithya, 2021) could not be found in PubMed; providing more complete reference might be helpful: Sunil, B. and Nithya, E. Correlation of Oxygen Saturation Index and Oxygenation Index in Hypoxaemic Respiratory Failure among Neonates. 2021 J Clin and Diagnostic Research, Vol. 15(8); DOI: 10.7860/JCDR/2021/49411.15237

Reviewer #2: This manuscript reports results from a prospective observational study comparing OSI and OI in neonates. I think the manuscript overall is well written, succinct, clearly understandable, and avoids extensive speculation. I do have a couple concerns and suggestions to improve the study.

One of my main concerns/suggestions relates to the fact that very similar studies have been done before, including the paper by Muniraman (PMID 30924897). I think that this (and any similar papers) need to be cited, and given that this similar research question has been studied already in this population, please talk about why this study needed to be done in the introduction and how this manuscript adds to the literature. I think this is a key thing that should be easy to address. You allude to the issue of limited resources in the introduction. The Muniraman study was retrospective so highlighting in the introduction and discussion the strength of your prospective approach would also be important.

Based on looking at other literature, it is probably worth adding in some subgroup comparison (maybe as supplemental figures) looking at term neonates and preterm neonates separately. Given age differences in fetal hemoglobin, these additional data could give important context and information for the readers.

In the methods, please provide an explanation as to why you chose your specific OI cutoffs? Were they data driven, based on decision thresholds for specific therapies, or some other reason.

Please provide additional information in the legend for Figure 2, specifically as it relates to the diagonal line included in your Bland-Altman plot; I don't think this is a component that most readers would be familiar with in this type of graph.

In the discussion, I'd suggest adding some comments about the resource-constrained setting. Examples of ideas that should be covered include why it is important to have/use a measure like OSI in lower-resourced settings. In high resource settings, these measures could drive use of things like nitric oxide, high-frequency ventilation, and ECMO. What is the utility for OSI (especially the higher value cutpoints) if advanced support modalities are not available? Please help the reader understand the importance.

Reviewer #3: The authors have done a great job in comparing the OI and OSI in infants with hypoxemic respiratory failure. I fully agree that the OSI could be of great value in neonatal care, especially in settings with less resources.

Although this paper is written clearly, I do have the following recommendations:

1. Please also include limitations in your manuscript. For example, elaborate on the influence of pH or temperature on the oxygen dissociation curve and thus on the association between OSI and OI.

2. To this reviewer it is not clear on which timepoints the OI/OSI are measured, please add this in your manuscript.

3. It is not explained whether preductal or postductal measurements are used. Did the authors think about this? Please elaborate on this in the paper.

4. The first paragraph of the discussion mainly repeats the introduction. My suggestion would be to remove this paragraph and to start the discussion with the results of your study.

5. I would suggest to add more, relevant, references to your sources as there have been many studies on this already.

6. Could the authors discuss whether the OSI should completely substitute the OI or whether the OSI and the OI should be used alongside each other?

Thank you in advance.

6. PLOS authors have the option to publish the peer review history of their article (what does this mean?). If published, this will include your full peer review and any attached files.

Reviewer #1: No

Reviewer #2: No

Reviewer #3: No

---

## [Author Response · Author response to Decision Letter 0]

27 Mar 2024

Manuscript ID number:

PONE-D-23-41595

Title of paper:

Assessing Hypoxic Respiratory Failure in Mechanically Ventilated Neonates: A Comparative Study of Oxygen Saturation Index and Oxygenation Index

RESPONSE TO REVIEWER’s COMMENTS

Reviewer #1:

1. Reviewer #1: Other investigators have also studied the correlation of OSI and OI in neonates, including those referenced in the manuscript, Rawat, et al., Sunit and Nithya. Similar articles not referenced include:

Muniraman, et al. (JAMA Network Open. 2019;2(3):e191179) and very recently

Maneenil, et al. (Am J Perinatol. 2024 Jan;41(2):180-186).

Given the similarities between the submitted article and the referenced and unreferenced articles, the authors need to point out if and how their results are similar or differ from others.

Reply: 

Thank you. We added these two references as your suggestion and expanded our discussion. Several previous studies were done for correlation of OSI and OI in ventilated neonates, there were some similar as well as differences among them. We pointed out that in the discussion part. 

Muniraman HK, Song AY, Ramanathan R, Fletcher KL, Kibe R, Ding L, Lakshmanan A, Biniwale M. Evaluation of oxygen saturation index compared with oxygenation index in neonates with hypoxemic respiratory failure. JAMA network open. 2019 Mar 1;2(3):e191179-.

Maneenil G, Premprat N, Janjindamai W, Dissaneevate S, Phatigomet M, Thatrimontrichai A. Correlation and prediction of oxygen index from oxygen saturation index in neonates with acute respiratory failure. American Journal of Perinatology. 2021 Nov 28.

2. It is unclear how many samples were obtained from the 123 enrolled patients. Were multiple samples obtained from some of the patients (as was the case for similar studies)? If not, how was the uniquely associated blood gas sample chosen?

Were arterial samples obtained via an indwelling arterial catheter (umbilical or peripheral) or by arterial stick? This is important to know as peripheral arterial sticks are more likely to have less reliability given the pain and discomfort associated with the blood draw.

Reply:

A total of 123 measurements from 123 neonates were analyzed. This means that only one pair of arterial blood gas (ABG) and corresponding oxygen saturation (SpO2) value was collected from each enrolled neonate. In our study, the ABG sample was drawn after the neonate had been on a ventilator for 30-60 minutes, which is a standard practice that allows for ventilator adjustments if needed. To minimize measurement errors, newborns were kept still during sampling, and no position changes, suctioning were performed during this time. SpO2 was continuously monitored using pulse oximetry. After observing a stable SpO2 value for at least one minute, the reading was recorded. ABG samples were obtained from umbilical arterial catheter or by arterial puncture. (The reason is that not all neonates had an indwelling arterial catheter at that time). 

We added the details above into the methods section to further clarify our study protocol. 

3. Comment:

Authors state on manuscript page 3: “It [OI] is widely utilized to guide clinical decisions, such as initiating high-frequency mechanical ventilation, administering inhaled nitric oxide therapy for neonates with respiratory failure caused by primary pulmonary hypertension, and assessing the response to surfactant therapy (3, 4).” Reference 3 should be updated to the 2017 Cochrane review by the same authors and others (Cochrane Database Syst Rev. 2017 Jan 5;1(1):CD000399). It should be noted that this reference only discusses the use of OI in the decision to use inhaled nitric oxide (INO). Reference 4 not being a manuscript publication warrants a website URL link. That reference discusses the use of OI for INO, possible transfer for ECMO, and to predict adverse outcome/death. Neither reference addresses the use of OI for high-frequency ventilation or response to surfactant.

Reply:

In the introduction, we removed references 3 and 4 and replaced them with 3 new references to support the sentence “ It is widely utilized to guide clinical decisions, such as administering inhaled nitric oxide therapy for neonates with respiratory failure caused by primary pulmonary hypertension, considering of high-frequency mechanical ventilation, and assessing the response to surfactant therapy” 

1./ Nelin LD, Potenziano JL. Inhaled nitric oxide for neonates with persistent pulmonary hypertension of the newborn in the CINRGI study: time to treatment response. BMC pediatrics. 2019 Dec;19:1-7.

2./ Amodeo I, Di Nardo M, Raffaeli G, Kamel S, Macchini F, Amodeo A, Mosca F, Cavallaro G. Neonatal respiratory and cardiac ECMO in Europe. European journal of pediatrics. 2021 Jun;180:1675-92.

3./Rong Z, Mo L, Pan R, Zhu X, Cheng H, Li M, Yan L, Lang Y, Zhu X, Chen L, Xia S. Bovine surfactant in the treatment of pneumonia-induced–neonatal acute respiratory distress syndrome (NARDS) in neonates beyond 34 weeks of gestation: a multicentre, randomized, assessor-blinded, placebo-controlled trial. European Journal of Pediatrics. 2021 Apr;180:1107-15.

4. Authors state on manuscript page 4: “Neonates with cyanotic congenital heart disease, hypotension, or in a state of shock were excluded…” It may be helpful to know how hypotension or shock were defined, especially considering that many babies with significant respiratory failure may have one or both, and for whom using OSI may be even more important. No such exclusion were used by Rawat, et al., Sunil and Nithya, Muniraman et al. or Maneenil, et al.

Reply:

As mentioned above, we obtained arterial blood gas (ABG) samples after neonates had been on a ventilator for 30-60 minutes. In cases of hypotension or shock, hemodynamic stabilization for neonates takes priority, and achieving stable blood flow in these neonates within a short timeframe can be challenging. Since reduced peripheral perfusion can lead to inaccurate SpO2 measurement from pulse oximetry, excluding neonates experiencing hypotension or shock during this initial period minimizes the impact of this interference factor on the value of SpO2.

In this study, hypotension is defined as mean blood pressure (MBP) below the normal limit in mmHg for a given gestational age and birth weight. Shock was confirmed or suspected in neonates exhibiting symptoms including lethargy, tachycardia or bradycardia, prolonged capillary filling time (>3 seconds), cold extremities, acrocyanosis, hypotension, narrow pulse pressure, urine output rate less than 1 mL/kg/hour, and increased serum lactate level greater than 2.8 mmol/L. (1,2)

(1) Nuntnarumit P, Yang W, Bada-Ellzey HS. Blood pressure measurements in the newborn. Clinics in perinatology. 1999 Dec 1;26(4):981-96.

(2) Sahni M, Jain S. Hypotension in neonates. NeoReviews. 2016 Oct 1;17(10):e579-89.

In addition, the accuracy of pulse oximetry for oxygen saturation (SpO2) varies considerably depending on the SpO2 range. Pulse oximeters are most accurate at higher saturation levels, with measurement accuracy declining significantly at lower SpO2 values (3). Therefore, we excluded neonates with cyanotic congenital heart diseases from this study. Muninaman et al also excluded infants with cyanotic congenital heart diseases in their study. 

(3) Ross PA, Newth CJ, Khemani RG. Accuracy of pulse oximetry in children. Pediatrics. 2014 Jan 1;133(1):22-9.

5. Minor editorial comments:

References in the manuscript should not use the authors first name or initial, e.g “Hadzic Devleta et al.” should read “Devleta et al.”. When there are only two authors, the referring phrase should be “Sunil and Nithya” instead of “Sunil et al.”

Reply: Thank you. We revised it as the correction of our mistakes.

6. One reference (Sunil and Nithya, 2021) could not be found in PubMed; providing more complete reference might be helpful: Sunil, B. and Nithya, E. Correlation of Oxygen Saturation Index and Oxygenation Index in Hypoxaemic Respiratory Failure among Neonates. 2021 J Clin and Diagnostic Research, Vol. 15(8); DOI: 10.7860/JCDR/2021/49411.15237

Reply:

Thank you. We cited this reference (No.9) as your suggestion to help readers easily assess this article. 

Reviewer #2: This manuscript reports results from a prospective observational study comparing OSI and OI in neonates. I think the manuscript overall is well written, succinct, clearly understandable, and avoids extensive speculation. I do have a couple concerns and suggestions to improve the study.

7. One of my main concerns/suggestions relates to the fact that very similar studies have been done before, including the paper by Muniraman (PMID 30924897). I think that this (and any similar papers) need to be cited, and given that this similar research question has been studied already in this population, please talk about why this study needed to be done in the introduction and how this manuscript adds to the literature. I think this is a key thing that should be easy to address. You allude to the issue of limited resources in the introduction. The Muniraman study was retrospective so highlighting in the introduction and discussion the strength of your prospective approach would also be important.

Reply: 

Thank you. We cited 2 new references that showed similar fields in research in the revised manuscript as your suggestion. 

• Muniraman HK, Song AY, Ramanathan R, Fletcher KL, Kibe R, Ding L, Lakshmanan A, Biniwale M. Evaluation of oxygen saturation index compared with oxygenation index in neonates with hypoxemic respiratory failure. JAMA network open. 2019 Mar 1;2(3):e191179-.

• Maneenil G, Premprat N, Janjindamai W, Dissaneevate S, Phatigomet M, Thatrimontrichai A. Correlation and prediction of oxygen index from oxygen saturation index in neonates with acute respiratory failure. American Journal of Perinatology. 2021 Nov 28.

We also added some information of cited references to show the similar and different among studies in introduction and discussion of the revised manuscript.

Unlike many previous studies on this topic, which were mainly retrospective (e.g., Devleta et al., Rawat et al., Muniraman et al., Maneenil et al.), our study is a prospective design. This approach guarantees meticulous data collection, especially in ensuring accurate pairing of SpO2 measurements with corresponding arterial blood gas samples. Furthermore, analyzing only one SpO2 and ABG pair per neonate fosters consistency in our study results compared to investigations that involve multiple measurements from the same neonates (Muniraman et al., Maneenil et al.). This decision is grounded in the fact that individual hemoglobin concentration can influence SpO2 readings. For instance, Muniraman et al. employed a median of 5 (range: 3-9) samples per patient, while Maneenil et al. utilized a median of 8 (range: 6-12) measurements.

8. Based on looking at other literature, it is probably worth adding in some subgroup comparison (maybe as supplemental figures) looking at term neonates and preterm neonates separately. Given age differences in fetal hemoglobin, these additional data could give important context and information for the readers.

Reply: 

Yes. We added preterm and term comparison in our revised manuscript as supplemental figure 1 (for preterm group) and Supplemental figure 2 (for term group) and added information of 2 subgroup comparison in result section.

9. In the methods, please provide an explanation as to why you chose your specific OI cutoffs? Were they data driven, based on decision thresholds for specific therapies, or some other reason.

Reply:

We selected OI cutoffs of >15 and >25 for analysis in this study. In general, OI cutoffs are widely used for categorizing the severity of hypoxic respiratory failure (HRF) in neonatal clinical practice. OI cutoffs are: ≤15: Mild HRF 16-25: Moderate HRF 26-40: Severe HRF 40: Very severe HRF. Supporting this choice, our data distribution reflects this categorization. Among the neonates enrolled, 103 (83.7%) had an OI ≤15, 16 (13.0%) had an OI between >15 and ≤25, only 4 (3.3%) had an OI between >25 and ≤40. Notably, none of the neonates presented with very severe condition (OI >40). Therefore, considering both established cutoffs and the distribution of OI values within our data, these specific cutoffs (>15 and >25) were chosen for data analysis.

We have added information to the methods section that explains why we chose certain OI cutoffs for this study.

10. Please provide additional information in the legend for Figure 2, specifically as it relates to the diagonal line included in your Bland-Altman plot; I don't think this is a component that most readers would be familiar with in this type of graph

Reply:

We have added in the legend for the figure. 

11. In the discussion, I'd suggest adding some comments about the resource-constrained setting. Examples of ideas that should be covered include why it is important to have/use a measure like OSI in lower-resourced settings. In high resource settings, these measures could drive use of things like nitric oxide, high-frequency ventilation, and ECMO. What is the utility for OSI (especially the higher value cutpoints) if advanced support modalities are not available? Please help the reader understand the importance.

Reply:

Thank you. We added the following paragraph in the discussion section to help the reader understand the importance of this study as your suggestion. 

OSI only requires pulse oximetry, which is widely accessible and available compared to the arterial blood gas analyzers needed for OI. Therefore, OSI has distinct advantage and become the preferred choice in resource-constrained setting where ABG analysis is limited or unavailable. Furthermore, even without access to advanced interventions like nitric oxide, high-frequency ventilation, or ECMO, higher OSI cutpoints can remain valuable in resource-constrained settings. They can help identify neonates at high risk of developing severe conditions, prompting closer monitoring and essential supportive care in treatment like oxygen therapy, infection control, and even referral to higher-level hospitals when possible. While these interventions may be less advanced, they can still significantly improve patient outcomes.

Reviewer #3: 

12. Please also include limitations in your manuscript. For example, elaborate on the influence of pH or temperature on the oxygen dissociation curve and thus on the association between OSI and OI.

Reply: 

The present study has some limitations. First, the data collection occurred at a single tertiary-level hospital in central Vietnam. While this limits the sample size and generalizability of the findings, it's important to note that this is the largest tertiary-level hospital in the region, specializing in the care of critically ill neonates. Additionally, it's the third largest hospital in Vietnam. Therefore, despite the sample size limitation, the data offers valuable insights as a pioneer study in this topic of research within Vietnam. Second, the relationship between PaO2 and SpO2 can be influenced by several factors, including pH, temperature, and PaCO2. These factors can cause a shift in the oxygen-hemoglobin dissociation curve, potentially complicating the analysis of the association between OSI and OI.

13. To this reviewer it is not clear on which timepoints the OI/OSI are measured, please add this in your manuscript.

Reply: 

In our study, the ABG sample was drawn after the neonate had been on a ventilator for 30-60 minutes, which is a standard practice that allows for ventilator adjustments if needed. To minimize measurement errors, newborns were kept still during sampling, and no position changes, suctioning, or other invasive procedures were performed during this time. SpO2 was continuously monitored using pulse oximetry. After observing a stable SpO2 value for at least one minute, the reading was recorded. ABG samples were obtained from indwelling arterial lines, either umbilical or peripheral.

We added the details above into the methods section to further clarify our study protocol. 

14. It is not explained whether preductal or postductal measurements are used. Did the authors think about this? Please elaborate on this in the paper.

Reply:

 Thank you. we added the following inform

---

## [Decision Letter · Decision Letter 1]

9 May 2024

Assessing Hypoxic Respiratory Failure in Mechanically Ventilated Neonates: A Comparative Study of Oxygen Saturation Index and Oxygenation Index

PONE-D-23-41595R1

Dear Dr. Nguyen,

We’re pleased to inform you that your manuscript has been judged scientifically suitable for publication and will be formally accepted for publication once it meets all outstanding technical requirements.

Kind regards,

David Chau

Academic Editor

PLOS ONE

Additional Editor Comments (optional):

Reviewers' comments:

Reviewer's Responses to Questions

**Comments to the Author**

1. If the authors have adequately addressed your comments raised in a previous round of review and you feel that this manuscript is now acceptable for publication, you may indicate that here to bypass the “Comments to the Author” section, enter your conflict of interest statement in the “Confidential to Editor” section, and submit your "Accept" recommendation.

Reviewer #1: All comments have been addressed

Reviewer #2: All comments have been addressed

Reviewer #3: All comments have been addressed

2. Is the manuscript technically sound, and do the data support the conclusions?

Reviewer #1: (No Response)

Reviewer #2: Yes

Reviewer #3: Yes

3. Has the statistical analysis been performed appropriately and rigorously? 

Reviewer #1: (No Response)

Reviewer #2: Yes

Reviewer #3: Yes

4. Have the authors made all data underlying the findings in their manuscript fully available?

Reviewer #1: (No Response)

Reviewer #2: Yes

Reviewer #3: Yes

5. Is the manuscript presented in an intelligible fashion and written in standard English?

Reviewer #1: (No Response)

Reviewer #2: Yes

Reviewer #3: Yes

6. Review Comments to the Author

Reviewer #1: The authors are commended for addressing the comments and feedback from the initial review, including the addition of more recent relevant journal articles.

Reviewer #2: I appreciate the revisions done by the authors. The revisions have addressed my initial concerns and I feel they have made this a much better manuscript.

Reviewer #3: (No Response)

7. PLOS authors have the option to publish the peer review history of their article (what does this mean?). If published, this will include your full peer review and any attached files.

Reviewer #1: No

Reviewer #2: **Yes: **Todd J Karsies MD, MPH

Reviewer #3: No

---

## [Editor Report · Acceptance letter]

17 May 2024

PONE-D-23-41595R1 

PLOS ONE

Dear Dr. Nguyen, 

I'm pleased to inform you that your manuscript has been deemed suitable for publication in PLOS ONE. Congratulations! Your manuscript is now being handed over to our production team.

Kind regards, 

on behalf of

Dr. David Chau 

Academic Editor

PLOS ONE